# Identifying and prioritising future interventions with stakeholders to improve paediatric urgent care pathways in Scotland, UK: a mixed-methods study

Emma King ![ORCID],[1] Emma France,[1] Cari Malcolm,[2] Simita Kumar,[3] Smita Dick,[4] Richard G Kyle,[5] Philip Wilson ![ORCID],[6] Lorna Aucott,[7] Stephen Turner ![ORCID],[8] Pat Hoddinott ![ORCID] [1]

**Correspondence to**
Dr Emma King;
emma.king@stir.ac.uk

## ABSTRACT

**Objectives** To identify and prioritise interventions, from the perspectives of parents and health professionals, which may be alternatives to current unscheduled paediatric urgent care pathways.

**Design** FLAMINGO (FLow of AdMissions in chIldren and youNG peOple) is a sequential mixed-methods study, with public and patient involvement (PPI) throughout. Data linkage for urgent admissions and three referral sources: emergency department, out of hours service and general practice, was followed by qualitative interviews with parents and professionals. Findings were presented and discussed at a stakeholder intervention prioritisation event.

**Setting** National Health Service in Scotland, UK.

**Participants** Quantitative data: children with urgent medical admission to hospital from 2015 to 2017. Qualitative interviews: parents and health professionals with experiences of urgent short stay hospital admissions of children. PPI engagement was conducted with nine parent–toddler groups and a university-based PPI advisory group. Stakeholder event: parents, health professionals and representatives from Scottish Government, academia, charities and PPI attended.

**Results** Data for 171 039 admissions which included 92 229 short stay admissions were analysed and 48 health professionals and 21 parents were interviewed. The stakeholder event included 7 parents, 12 health professionals and 28 other stakeholders. Analysis and synthesis of all data identified seven interventions which were prioritised at the stakeholder event: (1) addressing gaps in acute paediatric skills of health professionals working in community settings; (2) assessment and observation of acutely unwell children in community settings; (3) creation of holistic children's 'hubs'; (4) adoption of 'hospital at home' models; and three specialised care pathways for subgroups of children; (5) convulsions; (6) being aged <2 years old; and (7) wheeze/bronchiolitis. Stakeholders prioritised interventions 1, 2 and 3; these could be combined into a whole population intervention. Barriers to progressing these include resources, staffing and rurality.

## STRENGTHS AND LIMITATIONS OF THIS STUDY

⇒ Analysis of multiple sources of data from a robust mixed-methods study allowed us to identify interventions that stakeholders prioritised.
⇒ Health professional contributions represented most professional groups providing care to acutely unwell children.
⇒ Parents did not suggest specific interventions as solutions but shared their experiences and their values.
⇒ Children's and paramedics' views are not represented.
⇒ Solutions might differ for remote and rural communities and in a post-pandemic context.

**Conclusions** Health professionals and families want future interventions that are patient-centred, community-based and aligned to outcomes that matter to them.

## INTRODUCTION

Unscheduled hospital admissions of children in the UK have increased steadily over past decades, largely due to a rise in urgent short stay admissions (SSAs). For quantitative purposes, SSA can be defined as a patient being admitted and discharged on the same calendar day.[1 2] For qualitative sampling, since some parents cannot recall the precise time of admission and discharge, SSA can be defined as where parents' recall their child being admitted and discharged within 24 hours.[3] The rate of hospital medical admissions for children with acute illness in Scotland rose by 49% between 2000 and 2013, with SSAs rising by 186% from 8.6 to 24.6/1000 children per annum.[2] Children under 2 years of age account for the largest proportion of urgent SSAs,[1] with upper and

lower respiratory tract infections being a major reason for parents seeking an assessment.[2] Factors which influence decision-making leading to an admission (other than the child's well-being) include staff shortages, workload pressures, bed shortages, distance to hospital, local pathways of care and the family's social circumstances,[4] so changes to the present pathways of care are needed.[5]

Two recent systematic reviews have found that there is a weak evidence base to inform interventions aimed at safely avoiding acute admissions; there was limited evidence supporting the use of telemedicine, reconfiguration of staff and short stay admission units.[6 7] There is therefore a need for effective interventions to improve paediatric urgent care pathways to see if some admissions can be prevented[7] and improve family experiences.[3 4] It is not known which parts of the pathway interventions should target, for example, particular clinical settings or clinical presentations, or which interventions to develop.[7 8] The identification and development of complex interventions needs to take account of, and build on, existing evidence and be conducted in collaboration with patient and public involvement (PPI) and other stakeholders.[9 10]

The FLAMINGO project (FLow of AdMissions in chIldren and youNG peOple) is a sequential three phase mixed-methods study to investigate the pathway leading to urgent SSAs in children incorporating PPI throughout.[3] Phase 1, using linkage of national data sets, examined the pre-referral pathways for and characteristics of paediatric SSAs.[11] Phase 2 was informed by phase 1 findings and used qualitative interviews with parents and health professionals to explore contextual factors relating to SSAs, better understand referral pathways and develop priorities for future interventions aimed at improving unscheduled care pathways and the appropriateness of SSAs. In phase 3, an engagement event attended by phase 2 participants and wider stakeholders was held to share project findings and debate and prioritise interventions identified during the interviews.

This paper focuses on identifying and prioritising, from the perspectives of parents and health professionals, interventions which may improve the efficacy of current unscheduled urgent care pathways for children for future research and development.

## METHODS
### Study design
FLAMINGO is a three phase sequential mixed-methods study involving researchers from the National Health Service (NHS) and various universities in the UK. The research team included a collaboration of experts in quantitative and qualitative methodology and clinical practice. This paper describes the identification of potential interventions, informed by the IdentifyiNg and assessing different approaches to Developing compleX interventions (INDEX) approach (coauthor PH)[10] This systematic and consensus based intervention development guidance has not been previously applied to developing

interventions aiming to improve paediatric care pathways for SSA. It draws together findings from across the entire FLAMINGO study:

► PPI as a core component of the FLAMINGO project.[12 13]
► Phase 1: linked national data sets examining pre-referral pathways and characteristics of SSAs of children to Scottish hospitals. Methods are reported in online supplemental file 1 and in Dick *et al*[11]
► Phase 2: qualitative interviews with parents and health professionals providing insights into their values (including an important shared outcome of preserving the child's safety)[3] and experiences of unscheduled urgent care for children with SSAs and suggestions for change. Methods are reported in online supplemental file 1 and in Malcolm *et al*[3]
► In parallel a systematic review of hospital-based interventions to reduce acute paediatric admissions (coauthors SD, PW and ST) commenced at the start of the FLAMINGO study.[7] This was used to help identify interventions, along with data from the quantitative and qualitative research and PPI input. A second systematic review of primary and community care interventions to reduce urgent paediatric admissions had commenced during phase 2 but was not completed until after the intervention development work (coauthors SD, PW and ST).[6]
► Phase 3: a stakeholder event to discuss, debate and prioritise identified interventions for further research and development building on earlier PPI contributions.

FLAMINGO was undertaken in Scotland, where the NHS is organised into 14 geographically distinct Health Boards, each responsible for healthcare provision to their region's population. The project ran from January 2019 to December 2021 (figure 1). The FLAMINGO team met monthly, the quantitative and qualitative subteams met separately in between.

### Patient and public involvement
PPI was established at the outset (pre-COVID) to ensure the views of families were considered throughout all project stages. We involved nine parent–toddler groups and one university-based PPI advisory group attended by 112 adults and 107 children. Attendance at parent–toddler groups, including those accommodating people from lower socioeconomic groups and minority ethnic backgrounds and a university-based PPI advisory group, enabled parents to share their experiences of accessing healthcare and attending hospital for urgent healthcare. Their experiences informed the qualitative interview topic guide; ensured the materials used for recruitment were appropriate; and their experiences of urgent care supplemented the qualitative data to inform ideas for potential interventions. An independent PPI advisor critically reviewed and commented on manuscript drafts.

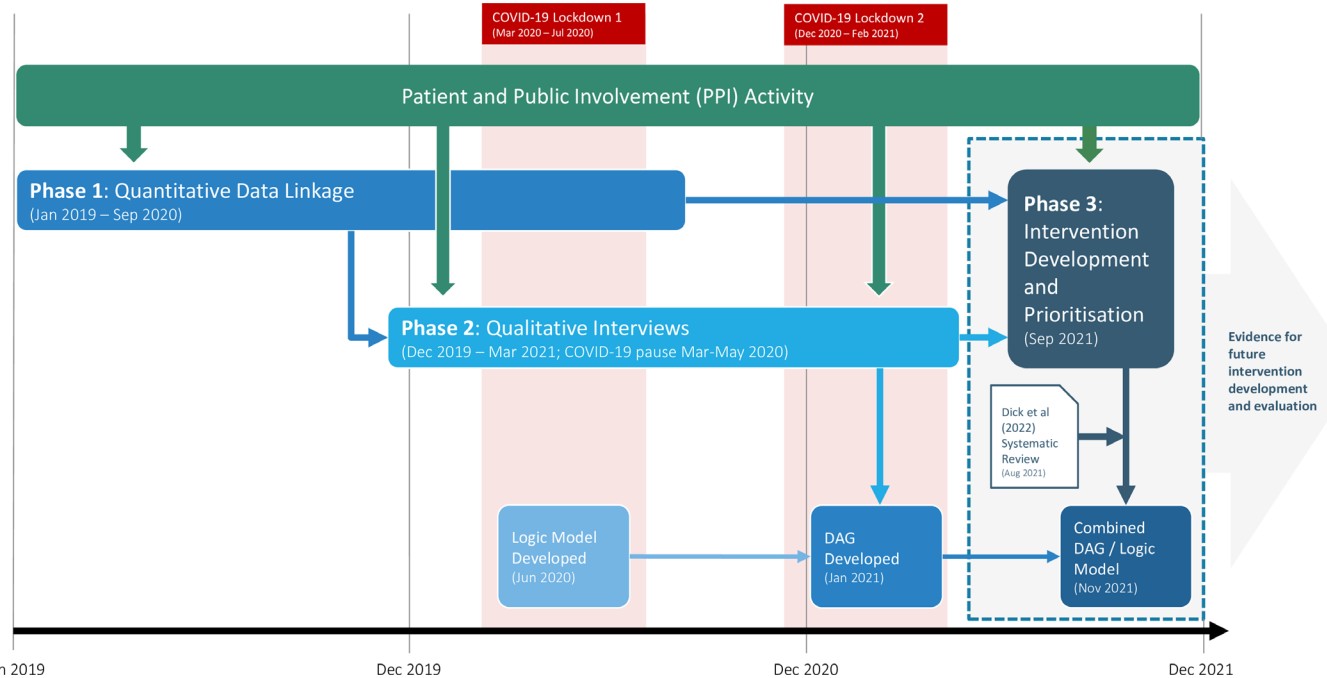

**Figure 1** Project timeline. DAG, directed acyclic graph.

## Intervention identification and development

### Data collection

The intervention development approach was target population-centred, incorporating front-line health professional and parent perspectives collected through PPI and qualitative interviews.[10] Potential future interventions were developed by the qualitative team (CM, EK, EF, PH) combining and interpreting FLAMINGO data, a systematic review[7] and the quantitative data showing the large contribution that wheeze/bronchitis and under 2's have to the number of paediatric SSAs.

Semi-structured interviews questions to health professionals asked about potential improvements to pathways for children between home and urgent SSAs; parents were asked about how their family's experiences could be improved in future. Interview topic guides are in online supplemental files 2 and 3 and Malcolm *et al.*[3] Participant information sheets are included in online supplemental file 4. For identification of interventions to take forward for prioritisation, an intervention was defined as a change/innovation where there is equipoise, that is, no evidence to support effectiveness, so it would need further research on acceptability, feasibility, effectiveness and cost-effectiveness before implementation.

A logic model was developed early in the study (figure 2) informed by a directed acyclic graph (DAG) model (a causal diagram).[14 15] The DAG was referred to iteratively throughout the FLAMINGO project and incorporated into a revised/refined logic model once the data analysis was complete (online supplemental files 5). The final logic model was informed by earlier FLAMINGO study qualitative analysis[3] about the shared outcomes of care that are important to both health professionals and parents that inform the design of a new care pathway:

prioritising child safety; resolving uncertainty and anxiety about the illness trajectory; parents greatly value timely access to care from experienced paediatric staff; and health professionals acknowledge the need to improve system pathways for prehospital care and support for families within the community.[3]

### Data analysis

Interventions were identified from the qualitative interview data through the following steps, guided by framework analysis for applied policy research[16] applied in QSR International NVivo V.12 software: familiarisation with transcripts; developing and agreeing a coding frame; indexing and further refinement of the coding framework; charting; and mapping and interpretation to search for patterns and explanations in the data. In-depth analysis was undertaken by EK and PH of potential solutions suggested by health professionals to improve urgent care pathways for children, any experiences of initiatives undertaken in their clinical settings, what had worked well and less well and any consequences together with drawing on the collective experiences of their professions. Similarly, in-depth analysis was undertaken of parents' accounts through both PPI consultation and interviews about how their family's experiences could be improved in future. Families mainly voiced their problems and experiences, therefore possible solutions were implicit, whereas Healthcare Professionals often suggested solutions explicitly. The team discussed the potential interventions generated through this process and decided whether each one met our intervention definition.

Reference was made to an underpinning systematic review of interventions to reduce acute paediatric hospital admissions, conducted by a FLAMINGO

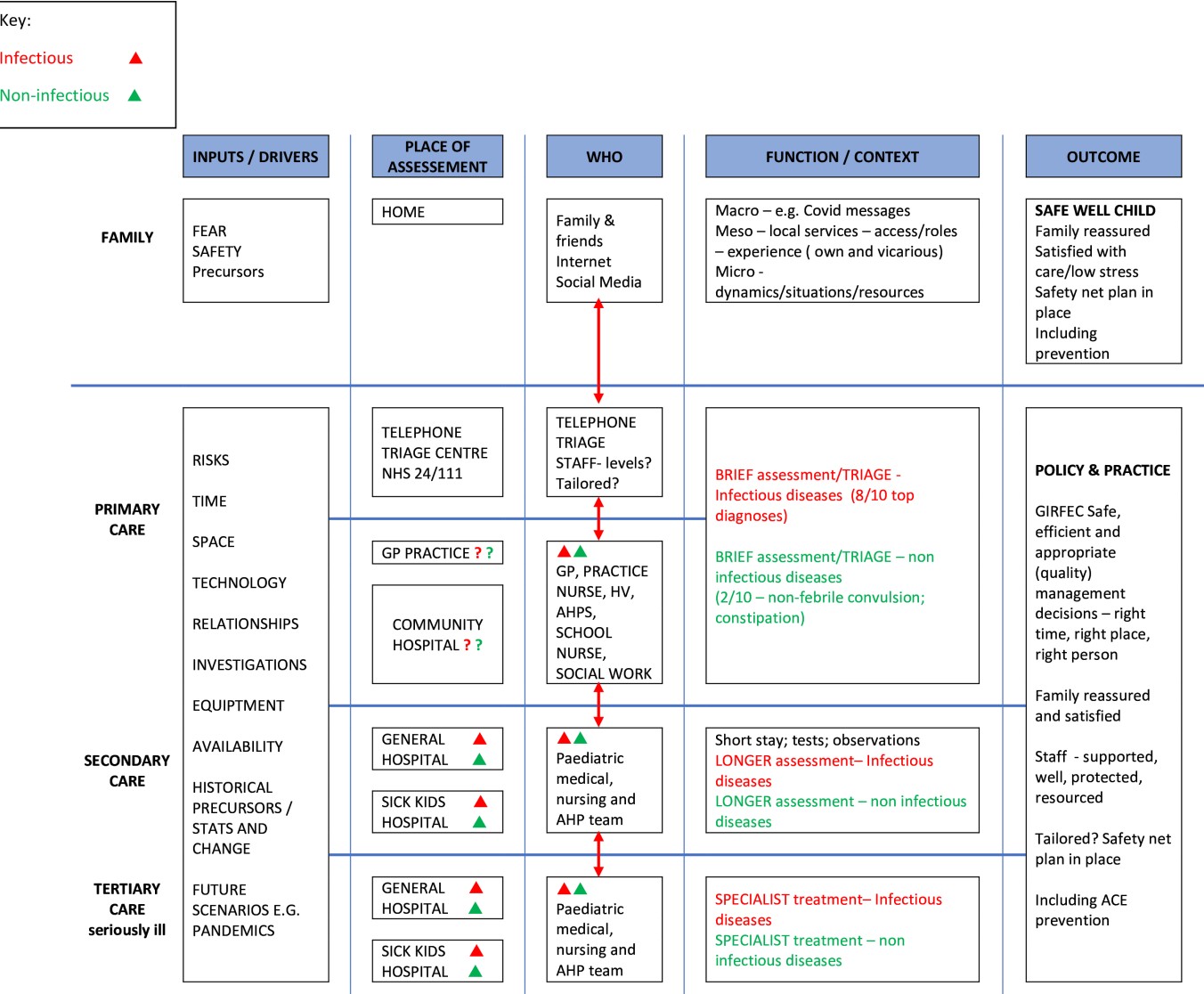

**Figure 2** Original system/process logic model—the acutely sick child. May 2020. AHP, Allied Health Professional; AHPS, Allied Health Professionals; GIRFEC, Getting It Right For Every Child; GP, general practice; HV, Health Visitor; NHS, National Health Service.

subteam, to generate a final list of interventions for stakeholders to prioritise.[7] The review findings were shared with the FLAMINGO qualitative team (EK, CM, EF, PH) once qualitative data collection was complete (table 1), although ST and PW contributed their knowledge gained from conducting the second systematic review in parallel with the qualitative data collection. The review of mostly hospital-based/secondary care interventions in 28 studies[7] identified four groups of interventions: condition specific care pathways; staff reconfiguration; new building; and telemedicine. Telemedicine interventions were ruled out of our list of interventions to prioritise, based on the qualitative data from professionals and parents; there had been a huge shift to using video/telephone consultation during the COVID-19 pandemic, therefore there was no longer equipoise. The review did not identify any additional interventions addressing the entire care pathway

from home to hospital which showed sufficient promise to pursue in our prioritisation.

### Intervention prioritisation
#### Data collection
We held an interactive stakeholder event to discuss and prioritise potential interventions on 3 September 2021. Attendees could attend in person or by video conference and included parents, health professionals and representatives from stakeholder organisations, for example, charities and parent groups, government. Six discussion groups with attendees were facilitated and audio-recorded by a team member (SD, EF, EK, ST, CM, RGK or PW) who also took written notes. Attendees individually ranked the priority of the interventions after the group discussions. Detailed methods are in online supplemental files 6.

**Table 1** Summarised quantitative and review data that informed intervention selection, triangulated with the qualitative data, for the stakeholder event

| Potential interventions identified from qualitative work | FLAMINGO data linkage results[11] King *et al* | Systematic review evidence[7] and other evidence sources |
|---|---|---|
| Addressing gaps in acute paediatric skills of health professionals working in community settings | Admissions of children with respiratory infections dominated SSAs from all referral sources indicating the need to improve skills and confidence in management of respiratory infections. Asthma, bronchiolitis, croup, lower respiratory tract infection and cough/wheeze/shortness of breath accounted for a total of 17 764 SSA (19% all SSA). Hypothesis: increasing paediatric skills of community staff in management of respiratory infections may benefit child outcomes and/or reduce SSA's. | Not addressed by the review. Supported by other studies showing increasing proportion of SSA 2000–2013;[2] a recognition that conditions may have been previously managed in the community;[2 5] knowledge that ~70% of GPs have no postgraduate paediatric training. |
| Assessment and observation of acutely unwell children in community settings | N/A | The review focused on interventions in secondary care only. |
| Creation of holistic children's 'hubs' | There were twice as many SSAs for children from the most deprived* compared with the least deprived communities (27% (n=25 022) vs 13% (n=12 032)). Those referred by ED (n=29 461) were over-represented by children from white ethnic groups 72.5% (n=21 360) compared with 56% from GPs and 63% from out of hours . Hypothesis: interventions in deprived communities may improve engagement with primary care; they may improve parent and child outcomes and/or reduce SSAs. | The review focused on interventions in secondary care only. Other evidence supports targeting more disadvantaged communities.[28] [29] |
| Adoption of 'hospital at home' models | Of all SSA's (n=92 229) n=12 378† readmissions within 30 days. Hypothesis: Hospital at home may improve parental and child outcomes; this may reduce the number of readmissions. | The review focused on interventions delivered in secondary care only. There is some evidence to support this.[30] |
| Extending specialised care pathways for convulsions | Ten composite diagnoses accounted for 52% (n=47 959) of SSA: asthma, bronchiolitis, convulsion (including febrile and afebrile convulsions), croup, gastroenteritis, upper respiratory tract infection, viral infection, tonsillitis, lower respiratory tract infections and admissions with a diagnosis of cough or wheeze or shortness of breath. Convulsions are most commonly admitted directly from ED, rather than through OOH or GP referral. Hypothesis: targeted interventions to the urgent care pathway that focus on specific diagnoses may improve parental and child outcomes and/or may reduce SSAs. | Evidence from the review was that care pathways for specific conditions are instrumental in reducing admissions especially when they are standardised. There was considerable heterogeneity between the studies and no randomised controlled trials were included. Publication bias was noted. |
| Extending specialised care pathways for children age <2 years old | Most SSAs were for children under 2 years—of the 92 229 children with an SSA, 44 063 (48%) were <2 years, 28 306 (31%) were <1. The median age for SSA was 2.2 years (interquartile values 0.74–5.8). | Not included in the review. Supported by other evidence.[31] |
| Extending specialised care pathways for wheeze/bronchiolitis | Prevalence of respiratory symptoms (15.6% of SSA were due to asthma, bronchiolitis and cough, wheeze and shortness of breath). | As described above. |

*Using Scottish Index of Multiple Deprivation for postcode of reported https://www.gov.scot/collections/scottish-index-of-multiple-deprivation-2020/.
†Not previously reported.
ED, emergency department; GP, general practice; OOH, out of hours service; SSAs, short stay admissions.

## Data analysis

Notes and transcripts from stakeholder group discussions were analysed in NVivo (by EF and EK) by coding data under 15 broad codes corresponding to confirming versus disconfirming perspectives for each of the interventions and a further code for other interventions suggested by attendees. Coded data were read repeatedly and themes identified. For the intervention priority ranking exercise, scores were summed and the average rank calculated for each of the identified interventions.

Participants in the qualitative interviews gave specific consent for their data to be used and anonymised quotes presented. All participants were provided with copies of the consent form by email in advance. Due to social distancing restrictions in place at the time written consent was replaced, where possible, by electronically signed consent forms and, where families did not have the facilities to do this, with verbal recorded consent given at the start of the interview. PPI and the stakeholder event did not require specific consent from individuals as they did not provide data. No voice recording was used for the PPI work. Those at the stakeholder event gave consent for voice recording for note taking purposes only, as such there are no quotes presented from the stakeholder event. Permissions for

the routinely collected data were given by the Public Benefit and Privacy Panel for Health and Social Care (reference 1718–0183). Data were analysed in the National Data Safe Haven of Scotland.[11]

## RESULTS
### Intervention identification and development
The results report how different evidence sources informed a short-list of interventions for prioritisation at the stakeholder workshop. Data were available from 92 229 SSA admissions, which were linked to other databases[11] and interviews with 48 health professionals (including general practices (GPs), hospital doctors and consultants, community nurses and hospital nurses) and 21 parents (including 20 mothers and 1 father of a child who had had an SSA for acute illness). Characteristics of participants are provided in online supplemental file 7 and elsewhere.[3]

### Identification of interventions
Analysis of all project data and a systematic review[7] identified seven potential interventions meeting our definition, four whole-system: (1) addressing gaps in acute paediatric skills of health professionals working in community settings, (2) assessment and observation of acutely unwell children in community settings, (3) creation of holistic children's 'hubs,' (4) adoption of 'hospital at home' models, and three subgroup interventions extending specialised care pathways for: (5) convulsions, (6) children age <2 years old and (7) wheeze, cough, shortness of breath/bronchiolitis. Table 1 summarises FLAMINGO data linkage results and systematic review data that support each intervention.

The first four suggested interventions align with reports published by the Scottish Government since the early 1970s which advocate care in the community.[17–19] The perspectives of professionals, parents and PPI supporting the interventions are described with selected illustrative quotations in box 1 and further quotations in online supplemental file 8.

### Addressing gaps in acute paediatric skills of health professionals working in community settings
Health professional interviewees described how staff working in the community often lack the necessary training and experience to manage acute paediatric conditions which could contribute to the increasing referrals of children to hospital. Three potential changes to the provision of care were identified: (1) increase specialist acute paediatric nursing roles, (2) provide additional paediatric training for GPs and (3) create rotational posts between primary and secondary care settings.

#### *Increase specialist acute paediatric nursing roles*
Professionals and parents perceive an assessment by a health professional with acute paediatric skills, at an early stage in the acutely unwell child's pathway, as beneficial.

---

**Box 1    Selected supporting interview quotations for each intervention**

**Addressing gaps in acute paediatric skills of health professionals working in community settings.**
**Increase specialist acute paediatric nursing roles**

It might be worth thinking about putting ANPPs in GP surgeries. Again, you've got well-experienced paediatric nurses that could go out into the community, see these patients, maybe be able to keep them at home by reassessing during the day, knowing what they're reassessing and also be able to do some teaching with the GPs. I think the way forward is maybe to try and put more paediatric experienced staff out in the community that can see acute unwell children. (C003_Nurse)

**Provide additional paediatric urgent care training for GPs**

Yeah, it's also unfortunate but true that paediatrics is not a requirement for training in general practice. So, you will unfortunately get, it's sad to say but the quality of referrals from primary care can be very poor. So, you have practitioners who are not prepared to take any chances themselves because of a lack of experience and this is a big problem for us in the winter months. (C014_Consultant)
Then she [GP] went to do his SATS, so she had one of those little probes, sort of like a completely mobile standalone thing and he wriggled and fussed […] she did take it several times although my concern was it was an adult probe, so I was thinking it's probably not particularly accurate, but anyway she did try several times, she was only getting 86. They [hospital] put a SATS monitor on him but for a small child and his SATS were like 99/100, so I think we were all like 'oh god, this seems such a waste of everyone's time and resources!' (Parent P010)

**Create rotational posts between primary and secondary care settings**

Part of my role when they took me on in post […] was going to be facilitation of learning and development of other people's skills. So, because COVID came in we like every other service had to set up a COVID assessment centre […] So for a wee while as the schools and nurseries all started to go back we had a bit of a boom in paediatric presentations and I actually came out of my out of hours post for a period of four weeks to support in-hours so that there was somebody there that could see the children, so that there was somebody there that could if the nurse practitioners wanted to come in and shadow. […] I suppose I'm quite lucky that I have got established relationships with the paediatric ward because that's where I came from. (C027_Nurse)
Yeah I'm a GP […]I also am piloting at the moment a joint clinic with a couple of the consultant paediatricians […]. We have a community child health consultant once a month and a general medical consultant once a month that we co-consult with on some more kind of challenging cases to try and prevent access issues and things that can potentially be dealt with in primary care. […]. (C031_GP)

**Assessment and observation over time in community settings**

I think the main thing with a lot of these kids is actually just time and giving them a chance to let their anti-pyretic settle and giving them a bit of fluid and just a period of observation, which I think is the limiting factor in GP practices and in GP out of hours, you know, they just don't have the facilities to be able to watch these children for a period of time. (C015_Doctor)

---

## Box 1   Continued

**Creation of holistic children's 'hubs'**

What I find is that then these children are born and there's nothing else for children and families apart from a health visitor and actually, you know, it's almost like if we could have hubs and community hubs where if you come in to get our health visiting weighed and things like that, you get taught about childhood diseases and when to worry, you know, and so almost like a mass public education programme that you get taught about when to worry, about when your child is sick. (C002_Consultant)

**Hospital at home model**

It's ebbs and flows and there are periods of times when you seem to get a number of referrals which you think, 'Surely that could be handled in the community, or can be managed in a different way rather than coming into hospital,' yeah. As to whether they could've handed in their urine sample of something, went away and then you can advise on what they're doing at home and representing, you know, safety-netting and so on. Or there are certain things where you think actually the best way to handle the particular scenario would've been to speak to someone who actually.[…] where a re-ferral is received at five o'clock in the evening or something along those lines, and you know that they need some investigation or im-aging that isn't going to happen that night […], actually, that child could potentially be risk assessed and managed at home and then referred to the appropriate services the next day. (C020_Consultant)

**Extend specialised care pathways for subgroups of children**

If they have a diagnosis of epilepsy and they've got a paediatric ep-ilepsy nurse specialist in their area I think it's pretty straightforward for them, they have a clear plan of what to do and who to contact. As I say, my job is to keep them out of hospital so in between clinic appointments they would be phoning myself if they have a seizure, they're advised to phone myself […]. (C041_Epilepsy Specialist Nurse)

Parents see specialist children's hospitals with a dedicated paediatric emergency department (ED) as a key advantage, with some criticising the lack of paediatric expertise in a general hospital ED. Health professionals recognise the potential benefits of increasing advanced nurse practitioners with specialist paediatric clinical assessment skills (ANPPs), within community settings and EDs not located in dedicated children's hospitals. ANPPs require the autonomy to act as senior decision-makers working with GPs who may facilitate the management of acutely unwell children at home safely and with appropriate referral to other services.

### *Provide additional paediatric urgent care training for GPs*

Paediatric training is not compulsory for GPs and exper-tise can vary across GPs and primary care staff. A knock-on effect is that newly-qualified GPs may be trained by GPs who have limited or no formal paediatric qualifications. Health professionals based in hospitals perceived that some children were referred to ED because GPs lacked the experience or confidence to provide care in the commu-nity. GPs who had undertaken additional paediatric

training acknowledged they had foregone training in other areas, such as care of the elderly or mental health.

Parental perspectives focused on comparing the reas-surance and expertise they felt when encountering staff in dedicated paediatric settings and on the barriers that some encountered in gaining this in the community.

GPs with experience and skills in acute paediatrics may be able to improve access, triage and appropriate referral of acutely unwell children in the community. Additional training for GPs may be feasible within the current UK 3-year GP training prior to certification to practice. In the longer-term this may improve urgent care pathways and reduce attendances at ED and/or SSAs.

### *Create rotational posts between primary and secondary care settings*

Health professionals indicate that staff rotating between primary and secondary care can build capacity, capability, knowledge, experience of illness trajectories and rela-tionships. Hospital staff gain greater awareness of issues such as the time pressures of 10 min appointment sched-ules in GP, and the concerns when a hospital paediatric service is hours away. Skills development is important, for example, ANPPs from hospitals working in primary care can support development of acute paediatric assessment skills among primary care staff.

Some health professionals identified gaps in the primary care workforce as a key issue for timely access to appropriate care, for example, where unfilled GP shifts in out of hours service (OOH) has led to increased ED work-load. While rotating staff between primary and secondary care may improve urgent care pathway outcomes, in the current staff resource context the feasibility of this is uncertain.

### Assessment and observation over time in community settings

A key outcome desired by parents and health professionals is to reduce uncertainty about the illness trajectory; this involves having time, space and staff to effectively assess and observe an unwell child.[3] However, hospital health professionals noted the lack of appropriate spaces for observation and availability of appropriately skilled staff to perform it in the community.

Health professionals reported a large proportion of SSAs to hospital were the result of children requiring an extended period of assessment and observation to inform clinical decision-making and safe discharge of the child home: for example, responses to a fluid challenge in the vomiting child, or to antipyretics in the febrile child.

Parents prioritised timely assessment of their acutely unwell child by a health professional, therefore, few deemed hospital attendances to be inconvenient. PPI views were largely consistent; however, parents did consider general ED, where both children and adults are waiting to be seen, as an inappropriate setting for chil-dren. The challenges, as reiterated by health professionals for providing child-friendly community settings, timely assessment and extended observation are: community

paediatric skills gaps and the current lack of infrastructure, equipment and suitable places, and the current NHS staff shortages.

The key overarching outcome of importance to both health professionals and families is safety of the child; parents value easy, direct access to urgent care for expert assessment and treatment.[3] Therefore, if community services were able to provide easy-access to staff with paediatric expertise and resources for timely assessment and observation, parents may be more likely to use community services than ED, with the potential to reduce SSAs to hospital.

### Creation of holistic children's 'hubs'
PPI indicated that parents have difficulties taking their child to the GP due to busy services and parental work. Parents in low paid or unstable work lose money taking time off to attend appointments and instead attend pharmacies and ED.

Health professionals see holistic children's 'hubs' as an alternative to increasing paediatric services within existing primary care systems, although significantly more resources and personnel would be required. They described potential hubs as providing a 'one stop shop' for children's health and psychosocial needs, such as health education, routine appointments, community paediatric appointments, social work advice and mental health services. Parents called for more tailored holistic care for both children and parents. Hubs could help by shifting more resources to community healthcare and integrating health visiting.

Some described hubs as combining the interventions discussed above: that is, staffed by professionals skilled in paediatrics who could observe and assess children. Children's hubs could be an accessible location, target disadvantaged communities and may improve outcomes and/or impact on GP and hospital workload, although in remote and rural areas there may still be considerable travel distance.

### Hospital at home model
Parent and toddler group PPI strongly indicated that families face multiple financial and logistical barriers to attending healthcare services including: no car; the high cost of public transport and taxis; no 24-hour public transport; parking challenges; time availability; and lack of childcare for other children.

Health professionals suggested an intervention which would provide more support and medical care for families in their own home, through a 'hospital at home' model. They cited hospital admissions for non-medical reasons, for example, parental concern, or lack of capacity in the current system to assess children or follow-up at home.

The experiences of some parents support the model, for example, a child with croup and parents' desire for easy and/or direct access to urgent care from paediatric specialists (depending on the intervention design and delivery). There is some complementarity and potential to combine interventions: increasing primary care staff paediatric skills, assessment and observation in the community, community hubs and hospital at home to address the outcomes of importance to parents and professionals.[8]

### Extend specialised care pathways for subgroups of children
Some parents described specialist care pathways for their children with acute or chronic relapsing conditions. Consultants, nurses and GPs spoke of the benefit of existing specialist care pathways for a minority of conditions, for example, epilepsy and diabetes. A primary aim of these pathways is to equip and support parents to manage children at home, and thus avoid unnecessary trips to hospital. Interviewees proposed extending specialist care pathways to other acute conditions and/or age groups as a way of triaging to appropriate expertise.

Many health professionals viewed babies with fever, and young children with respiratory illnesses as primary targets for future interventions which may improve their outcomes and impact on attendances in primary and secondary care. Convulsions were frequently raised and are a common reason for SSA explored in depth elsewhere (Malcolm *et al*, 2023).

### Results of stakeholder intervention prioritisation
The stakeholder event had 47 attendees (22 attending online): 7 parents, 12 health professionals (primary and secondary care nurses, consultants, nurse consultants and GPs) and 28 representatives from stakeholder organisations (charities and parent groups' coordinators, academics and government).

### Increase community specialist acute paediatric nursing roles for care of the acutely unwell child
Stakeholders affirmed the importance of ANPPs with expertise in care of the acutely unwell child, especially in areas where patients live far from a hospital. A skilled ANPP in the GP practice could potentially send parents home with a care plan and prevent SSAs and/or reduce pressure on OOH services. ANPPs are seen as appropriate professionals to undertake neonatal assessments and manage chronic paediatric conditions for which children would otherwise attend hospital outpatient clinics. A caveat expressed by stakeholders (but not interviewees), was the importance of ANPPs being rooted in secondary care, whether through rotation or regular shifts, to keep up to date and avoid losing their skills in dealing with acutely unwell children. More integrated and fluid professional roles across primary–secondary care services, rather than the current dichotomy of community versus hospital were envisioned.

Stakeholders identified several challenges for increasing community ANPP roles: to fully train an ANPP can take around 5 years, so increasing numbers needs advance planning; some hospital ANPPs do not see children under the ages of 5 or 12 years, and thus require more training to become confident in managing younger

children; many community ANPP roles are advertised for adult care only. Some hospital health professionals lamented the potential loss of their skilled ANPPs to primary care. Stakeholders with paediatric nursing experience suggested extending roles in primary care could deskill other GP and nursing staff through reducing contact with unwell children.

### Improve paediatric skills by creating rotational posts between primary and secondary care settings

Stakeholders recognised the value of observing acutely unwell children in community settings but acknowledged that it would require confidence-building in both staff and parents. Increasing paediatric skills in the community for GPs and nurses was acknowledged to be dependent on population geography, existing service structure and staff resource, with only one health professional specifically mentioning rotational posts for nurses. Others mentioned the benefits of less formal crossovers between care settings, for example, skills gained from previous roles, GPs training in secondary care or hospital clinicians holding occasional clinics in primary care. A more integrated/fluid rotation of staff across primary and secondary care would provide a 'safety net' when creating a service that relies on a few highly specialised staff.

### Creation of holistic children's 'hubs'

Similar to interviewees, stakeholders suggested potential hubs as providing holistic paediatric services include a focus on preventing and managing childhood illnesses. The hubs would be situated in the community with good transport links and parking and could provide care for extended hours, bridging the current gap between GP opening hours (08:00 to 18:00) and the busiest time in urgent care (18:00 to 02:00). A few health professionals commented that observation and assessment in hubs may particularly suit children where National Institute of Health and Care Excellence (NICE) pathways apply, for example, bronchiolitis.

Unlike interviewees, stakeholders raised several concerns about potential hubs: they may become too busy; parents may become over-reliant rather than learning how to manage childhood illnesses at home; information and advice currently provided by the 24-hour NHS telephone advice and triage service (NHS 24) could be replaced by the hubs, which may result in unforeseen issues such as a lack of access to the large bank of language interpreters. A substantial cultural shift in parental confidence with primary care and community services will be required, otherwise parents may simply bypass the hubs and go straight to ED.

Stakeholders perceived the massive investment and political change required for hubs as unrealistic, given previous promises of more investment in community health services have not translated into practice. Furthermore, they highlighted that interventions such as hubs would only be viable with adequate staffing which could be a barrier with the current workforce shortage.

### Hospital at home model

Stakeholders discussed the hospital at home model only briefly. Some stakeholders raised concerns about who would staff a hospital at home model because this might deplete staff in existing services, putting even more strain on them. Others felt that hospitals at home might be a way of retaining staff, for example, nurses, who no longer want to work on acute wards. In support of the model, some parents stated that, with hindsight, the help and reassurance they had received from SSAs could have been provided in the community and the model could potentially provide the continuity of care and familiar staff that parents want.

### Extend specialised care pathways

Stakeholders spent less time discussing specialist care pathways for bronchiolitis, convulsions and infants aged under 2 years. Some parents had experience of specialist diabetic and epilepsy nurses and praised the care their children had received. Health professionals preferred condition-based, rather than age-based, pathways. They felt that specialist nursing was appropriate for relatively uncommon conditions but was not suitable for frequently occurring conditions and those with existing standardised NICE pathways, such as bronchiolitis. Even where NICE care pathways do exist, health professionals lamented that adherence is variable and education is needed. They recognised that if a child arrives with multiple issues, it often leads to health professionals' confusion over which pathway to follow.

Other specialised pathways were discussed, for example, a paediatric-specific NHS 24 phone line, but it was concluded this might be confusing for parents to have yet another telephone number and that staffing it would be challenging. Stakeholders wanted greater recognition of direct referral to prehospital specialists, such as opticians (NHS 24 already refer children straight to opticians when appropriate) and pharmacists, who could advise parents around common childhood illnesses.

### Results of ranking the priority of interventions

Twenty attendees at the interactive stakeholder event (16 in person and 4 online) submitted their anonymised priority rankings for the interventions they considered most important with a score of 1 indicating the most popular option and 7 the least popular option. Table 2 shows a cluster of three more popular options (ranks 1–3) and options that were not so popular (ranks 4–7). Stakeholders considered creating specialist care pathways for bronchiolitis, convulsions and infants aged under 2 years as lower priority interventions, with the lowest priority pathway being one for infants.

## DISCUSSION

In a robust mixed-methods study combining analysis of quantitative and qualitative data, PPI engagement and systematic review evidence,[7] we were able to identify and

**Table 2** Results of stakeholder intervention prioritisation ranking

| Potential intervention | Mean score | Rank |
|---|---|---|
| Addressing gaps in acute paediatric skills of health professionals working in community settings. | 2.53 | 1 |
| Creation of holistic children's 'hubs'. | 2.73 | 2 |
| Assessment and observation of acutely unwell children in community settings. | 3.00 | 3 |
| Hospital at home model. | 4.47 | 4 |
| Specialised care pathways—for wheeze/bronchiolitis. | 4.76 | 5 |
| Specialised care pathways—for convulsions. | 4.94 | 6 |
| Specialised care pathways—for under 2s. | 5.33 | 7 |

prioritise interventions with potential to improve urgent care pathways for children. Our results can inform more efficient care pathways to improve parent experiences when seeking care for acutely sick children and potentially reduce or prevent hospital admissions. Of seven potential interventions, stakeholders, including parents and health professionals, identified three higher priority and four lower priority future interventions. Higher priority interventions were to address gaps in the acute paediatric skills of health professionals working in community settings, to create holistic children's hubs and to facilitate assessment and observation of acutely ill children in community settings. Lower priority interventions were 'hospital at home' and specialised care pathways for convulsions, infants under 2 years old and wheeze/bronchiolitis. The higher priority interventions could be combined into a whole population intervention which provides community-based assessment and observation by appropriately skilled health professionals in an accessible location/hub.

Strengths include using the INDEX guidance[10] for intervention development which has not been followed previously to address the gap in the systematic evidence[7] about how to improve prehospital acute paediatric care and combining qualitative and quantitative methods with stakeholder engagement and PPI. Different data collection, settings and framing of approach generate different perspectives and help in the search for disconfirming data. The health professional contributions are representative of different professional groups (apart from paramedics who we tried unsuccessfully to recruit) and represented a range of urban and rural/remote areas.

Limitations include that parents talked about their care experiences and preferences, but did not offer specific solutions. Due to COVID-19, we could not recruit parents and children face-to-face at hospitals as intended which limited sample diversity, although our spread of deprivation measured by Scottish Index of Multiple Deprivation was good. Children's views are not represented, as many were infants and no-one took up the invitation of a video conference or telephone interview with their child. Solutions for cities might look different to those for remote and rural communities which may have been underrepresented at the stakeholder event. Rural and remote areas have distance, travel and different workload/

resource considerations. Generalisability of findings beyond Scotland and to the post-COVID years is uncertain, however many staff have cross-UK and international experience. The priority ranking exercise was exploratory and only 20 out of 47 attendees completed it.

Our study is unique in its inclusion of health professional perspectives and triangulation of different stakeholder perspectives to inform paediatric care intervention development. Previous published qualitative research has focused on parents' perspectives of admissions.[20 21] Prioritising what matters to patients and health professionals (the target population) is important when designing an intervention with a congruent theory of change and will inform future decisions about the focus for future evaluation design.[22] Our findings complement a recent rapid literature review of (not paediatric-specific) patient urgent and emergency care experiences.[23] Another recent review of the literature was published after our study was completed.[6] This second review found that telemedicine was useful in terms of preventing admissions from the community, while the initial systematic review had focused on hospital-based interventions. Although this did not inform our findings it also calls for prompt introduction and evaluation of improved prehospital pathways for the care of acutely unwell children.

The importance of rotation of staff and the role of ANPPs in integrated services to facilitate early referral to the Community Children's Nursing Teams as an alternative to hospital care; strong personal relationships between consultants and community children's nurses are considered important.[24] In England, what reassures professionals, especially GPs, when referring to Community Children's Nurses are clear clinical governance protocols.[25] There is some evidence to indicate that an experienced GP with paediatric training may be the best person to decide whether a child is ill or not.[26] Research in Scotland revealed that the quality of relationships, communication and expectations between GPs and hospital specialists at the interface between primary and secondary care were important influences on patient care.[27]

Evidence of the best short-term and long-term outcomes for children and families is required before novel pathways are implemented into routine acute care pathways. Further intervention acceptability, feasibility/pilot testing

research is required using randomised controlled trial methodology to establish effectiveness, cost-effectiveness and any unintended consequences over short-term to longer-term horizons.

The key message for policymakers is that to improve paediatric urgent care pathways stakeholders, including health professionals and families, want future interventions that are safe, patient-centred and community-based.[3] Close collaboration between academics and policymakers, senior decision-makers and potential funders will be required to ensure that future interventions value contributions from parents and front-line health professionals and are rigorously evaluated to advance evidence-based policy and care that improves child health outcomes.

**Author affiliations**
[1]Nursing, Midwifery and Allied Health Professions Research Unit, University of Stirling, Stirling, UK
[2]School of Health Sciences, University of Dundee, Dundee, UK
[3]Screening and Immunisation, Public Health Scotland, Edinburgh, UK
[4]Child Health, University of Aberdeen, Aberdeen, UK
[5]Academy of Nursing, Department of Health and Care Professions, Faculty of Health and Life Sciences, University of Exeter, Exeter, UK
[6]Institute of Health and Wellbeing, University of Aberdeen, Aberdeen, UK
[7]Centre for Randomised Healthcare Trials, University of Aberdeen, Aberdeen, UK
[8]Women and Children Division NHS Grampian, Aberdeen, UK

**Acknowledgements**  Amy Woodhouse of Children in Scotland registered charity critically reviewed and commented on drafts of this manuscript. We are grateful for Dave Kelly at Albasoft for providing primary care data. We are grateful to Rebecca Fairnie at Electronic Data Research and Innovation Service for managing our access to all data. The authors would like to extend our thanks to the parents and health professionals who participated in this study, and to the stakeholders who participated in both public and patient involvement activities and the engagement and prioritisation event.

**Contributors**  CM: Conceptualisation, Formal analysis, Funding acquisition, Investigation, Methodology, Project administration, Writing—original draft. EK: Data curation, Formal analysis, Investigation, Writing—review and editing. EF: Conceptualisation, Formal analysis, Methodology, Writing—review and editing. RGK: Conceptualisation, Funding acquisition, Methodology, Writing—review and editing. SK: Methodology, Writing—review and editing. SD: Project administration, Writing—review and editing. PW: Conceptualisation, Funding acquisition, Methodology, Writing—review and editing. LA: Conceptualisation, Methodology, Writing—review and editing. ST: Conceptualisation, Funding acquisition, Methodology, Project administration, Supervision, Writing—review and editing, Guarantor of the study. PH: Conceptualisation, Formal analysis, Funding acquisition, Investigation, Methodology, Validation, Writing—review and editing.

**Funding**  This work was supported by the Chief Scientist Office of the Scottish Government, grant number HIPS/18/09.

**Competing interests**  None declared.

**Patient and public involvement**  Patients and/or the public were involved in the design, or conduct, or reporting, or dissemination plans of this research. Refer to the Methods section for further details.

**Patient consent for publication**  Not applicable.

**Ethics approval**  This study involves human participants and was approved by North of Scotland Research Ethics Service 19/NS/0134. Participants gave informed consent to participate in the study before taking part.

**Provenance and peer review**  Not commissioned; externally peer reviewed.

**Data availability statement**  Data may be obtained from a third party and are not publicly available. Data are available upon reasonable request. Quantitative data can be obtained on approach to Public Health Scotland. Qualitative data can be shared on reasonable approach to the authors.

**ORCID iDs**
Emma King http://orcid.org/0000-0003-3611-9647
Philip Wilson http://orcid.org/0000-0002-4123-8248
Stephen Turner http://orcid.org/0000-0001-8393-5060
Pat Hoddinott http://orcid.org/0000-0002-4372-9681

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
