## [Reviewer comments · BMJ Open]

ARTICLE DETAILS

TITLE (PROVISIONAL)	Identifying and prioritising future interventions with stakeholders to improve paediatric urgent care pathways in Scotland, UK: a mixed-methods study
AUTHORS	King, Emma; France, Emma; Malcolm, Cari; Kumar, Simita; Dick, Smita; Kyle, Richard G; Wilson, Philip; Aucott, Lorna; Turner, Stephen; Hoddinott, Pat

VERSION 1 – REVIEW

REVIEWER	Hooft, Anneka University of California San Francisco, Emergency Medicine
REVIEW RETURNED	19-Jun-2023

GENERAL COMMENTS	This is a large, very detailed study on potential interventions for the increasing number of short-stay pediatric admissions in Scotland. The qualitative results were very strong and well-explained, and the discussion tied this all together well. My main concern was that the methodology was difficult to follow, particularly with this being part of a larger study and how it was organized. Adding clarity to the methods would make this stronger and easier to follow. Abstract: FLAMINGO is referenced in the abstract Design section without any context. Is this the name of the current study? Is this a parent study? It may help to make the title a direct reference to FLAMINGO with a subtitle for this specific phase of the study. Intro: Please define short stay admission, though this may be a defined entity in the UK, as a non-UK-based provider, I was not sure the timeframe this describes. Line 16: Some specifics in lieu of the word “pressures” would aid in understanding the challenges faced and describe the context a bit better. Listing some of these contributors such as high patient volumes, limited facilities or access, limited numbers of hospital beds, etc. Line 25: what types of interventions were these that failed? Hospital-based? Public health? Both? It would also help to transition this into the next line of why this indicates that urgent care pathways need to be improved. Is this because we cannot prevent admissions, so we should streamline the urgent care process? Or, because urgent care systems may still be able to prevent admissions? The end of this paragraph could set up the need for this study a little more and state the hypothesis more clearly. I understand this is a more exploratory study but emphasizing the need for this study more clearly would add impact. It was a little unclear to me what is being studied by the end of the intro section. Methods:
---

	Again, is FLAMINGO the parent study? Is this study part of FLAMINGO? Would set up the overarching structure a bit more to understand how this fits. Also, if INDEX is an acronym, this also should be defined. The initial summary of the team’s work I believe relates to the same components described in the methods. If this is the case, the order should be the same. E.g. PPI is listed on line 15 and the 3rd component of the prior work, but then is the first subheading of the methods section. This would make it easier to follow which component of the study is being referred to and the temporal order. This is laid out more clearly in the results and there is a later a reference to “Phase 2.” It would help to organize the methods similarly so it is easier to understand the methods step-wise. Perhaps also describing the study as explanatory sequential, exploratory sequential, or parallel mixed methods would help me understand which portion came first. Line 53: What does this mean that the INDEX approach has not been followed previously. Followed by whom? To address which problem? Data Analysis: The first paragraph here seems to belong more in the results? Or are these previously identified themes that were used to guide the analysis? Figures are not labeled nor do they have any sort of legend Results: Table 1 seems to be structured with the interview/qualitative suggestion/theme then the support from data linkage and review. Would help to label column 1 as such. Table 2 is very helpful. The results layout is clearer than the methods as divided in to distinct “Phases” of the study. Would help to make the methods section align with this. Discussion: In table 1, it is described that deprived/disadvantaged communities may have more admissions and require more targeted intervention. It is unclear how well interview participants or stakeholders represented this group and what is considered “deprived.” I assume that was defined in the prior studies referenced in the table but would help to know if those same characteristics were well-represented among qualitative participants. Again, here in the discussion it is unclear to me whether these interventions would “improve urgent care pathways” or if the ultimate outcome is to reduce unnecessary admissions by improving primary care and outpatient management of children. I was not sure where urgent care falls into this process. Again, some of this may be due to my limited familiarity with NHS.
--	--

REVIEWER	Sayegh, Caitlin Children's Hospital Los Angeles
REVIEW RETURNED	26-Jun-2023

GENERAL COMMENTS	This manuscript is clear and useful. The comprehensive supplementary files are helpful to supporting others in replicating this work or applying similar methods. Could the authors more explicitly discuss in the body of the manuscript which participants provided informed consent (e.g., qualitative interviews, health boards, stakeholders?) and how? Was the study overseen by an ethics review board? Some of this
---

	information could be inferred in supplementary files but should be summarized clearly in methods section.
--	---

VERSION 1 – AUTHOR RESPONSE

Reviewer: 1

Dr. Anneka Hooft, University of California San Francisco

Comments to the Author:

This is a large, very detailed study on potential interventions for the increasing number of short-stay pediatric admissions in Scotland. The qualitative results were very strong and well-explained, and the discussion tied this all together well. My main concern was that the methodology was difficult to follow, particularly with this being part of a larger study and how it was organized. Adding clarity to the methods would make this stronger and easier to follow.

Abstract: FLAMINGO is referenced in the abstract Design section without any context. Is this the name of the current study? Is this a parent study? It may help to make the title a direct reference to FLAMINGO with a subtitle for this specific phase of the study.

FLAMINGO is a sequential mixed-methods study with data-linkage followed by qualitative interviews and then the stakeholder intervention priority event. We clarified this in the abstract and throughout the paper.

Intro:

Please define short stay admission, though this may be a defined entity in the UK, as a non-UK-based provider, I was not sure the timeframe this describes.

We have now defined this in the introduction.

Line 16: Some specifics in lieu of the word “pressures” would aid in understanding the challenges faced and describe the context a bit better. Listing some of these contributors such as high patient volumes, limited facilities or access, limited numbers of hospital beds, etc.

We have edited the text to make it clearer what pressures we are discussing.

Line 25: what types of interventions were these that failed? Hospital-based? Public health? Both? It would also help to transition this into the next line of why this indicates that urgent care pathways need to be improved. Is this because we cannot prevent admissions, so we should streamline the urgent care process? Or, because urgent care systems may still be able to prevent admissions?

We have edited the text to make it clearer which types of interventions were discussed in two recent systematic reviews and that urgent care pathways need to be improved to both prevent admissions and improve family experiences.

The end of this paragraph could set up the need for this study a little more and state the hypothesis

more clearly. I understand this is a more exploratory study but emphasizing the need for this study more clearly would add impact. It was a little unclear to me what is being studied by the end of the intro section.

We have included a more detailed description of the wider FLAMINGO study, emphasised the need for interventions to see if SSA can be prevented given the limited systematic review evidence and to improve parental experiences of care for their acutely sick child

Methods:

Again, is FLAMINGO the parent study? Is this study part of FLAMINGO? Would set up the overarching structure a bit more to understand how this fits.

This has been clarified as above in the abstract and introduction. Phases are added to the bullet points summarising the methods to make the overall study design clearer.

Also, if INDEX is an acronym, this also should be defined.

INDEX is now defined.

The initial summary of the team's work I believe relates to the same components described in the methods. If this is the case, the order should be the same. E.g. PPI is listed on line 15 and the 3rd component of the prior work, but then is the first subheading of the methods section. This would make it easier to follow which component of the study is being referred to and the temporal order. This is laid out more clearly in the results and there is a later a reference to "Phase 2." It would help to organize the methods similarly so it is easier to understand the methods step-wise. Perhaps also describing the study as explanatory sequential, exploratory sequential, or parallel mixed methods would help me understand which portion came first.

We have changed the order of the PPI in the bullet point summary of methods as it was integral to the study from beginning to end and added. We have added Phases to the bullet pointed list and clarified that the systematic reviews were undertaken in parallel to the entire FLAMINGO study. We have removed the later reference to 'Phase 2' in the results as we agree with the reviewer this is confusing to the reader. Sequential has been inserted before mixed methods in the abstract and methods. We have not included the term explanatory as the data linkage included some exploratory analyses. Line 53: What does this mean that the INDEX approach has not been followed previously. Followed by whom? To address which problem?

We have defined the INDEX acronym and explained that this systematic and consensus based intervention development guidance has not been previously applied to developing interventions aiming to improve paediatric care pathways for SSA.

Data Analysis: The first paragraph here seems to belong more in the results? Or are these previously identified themes that were used to guide the analysis?

We agree that this is better placed in the previous section as these are findings of earlier qualitative analysis from the FLAMINGO study that contributed to revisions of the logic model and informed the intervention development process. We have moved this paragraph to the previous section.

Figures are not labelled nor do they have any sort of legend

We would welcome the editorial team guidance on this. Figure legends and labels are included in their appropriate place in the text, and on the file labels but not in the supplementary files, as is our

interpretation of the author formatting requirements. We have duplicated at the bottom of the file for completeness.

Results:

Table 1 seems to be structured with the interview/qualitative suggestion/theme then the support from data linkage and review. Would help to label column 1 as such.

Relabelled as suggested

Table 2 is very helpful.

The results layout is clearer than the methods as divided in to distinct “Phases” of the study. Would help to make the methods section align with this.

We have addressed this in the introduction, the methods and the results, in particular by adding Phases to the bullet point summary in the methods and identifying that the systematic reviews were undertaken in parallel and sequentially. As the INDEX intervention development approach draws on all of the FLAMINGO dataset, we believe it makes sense to have the ‘Intervention development’ and ‘Intervention prioritisation’ sections. We have edited headings to make the alignment clearer.

Discussion:

In table 1, it is described that deprived/disadvantaged communities may have more admissions and require more targeted intervention. It is unclear how well interview participants or stakeholders represented this group and what is considered “deprived.” I assume that was defined in the prior studies referenced in the table but would help to know if those same characteristics were well-represented among qualitative participants.

Using Scottish Index of Multiple Deprivation for postcode of residence is a robust methodology for ensuring deprivation is comprehensively reported for the data linkage results of the Flamingo study (Table 1). We have added a reference to this database as a footnote to Table 1. SIMD by postcode was collected for parents participating in interviews and has been added to the characteristics table in Supplementary file 7.

Again, here in the discussion it is unclear to me whether these interventions would “improve urgent care pathways” or if the ultimate outcome is to reduce unnecessary admissions by improving primary care and outpatient management of children. I was not sure where urgent care falls into this process. Again, some of this may be due to my limited familiarity with NHS.

A sentence has been added to the discussion clarifying that our results can inform more efficient care pathways to improve parent experiences when seeking care for acutely sick children and potentially reduce or prevent hospital admissions

Reviewer: 2

Dr. Caitlin Sayegh, Children's Hospital Los Angeles

Comments to the Author:

This manuscript is clear and useful. The comprehensive supplementary files are helpful to supporting others in replicating this work or applying similar methods.

Could the authors more explicitly discuss in the body of the manuscript which participants provided informed consent (e.g., qualitative interviews, health boards, stakeholders?) and how? Was the study

overseen by an ethics review board? Some of this information could be inferred in supplementary files but should be summarized clearly in methods section.

The project was overseen by NHS research ethics. This information has been moved from its current place in the 'study design' section to a new 'Ethics' section at the bottom of the methods. The specific consent from participants has also now been included.

Reviewer: 1

Competing interests of Reviewer: none

Reviewer: 2

Competing interests of Reviewer: No competing interests.